# CheXmask-U: Quantifying uncertainty in landmark-based anatomical segmentation for X-ray images

Matias Cosarinsky[1,2]             MCOSARINSKY@DC.UBA.AR
Nicolas Gaggion[1,3]               NGAGGION@DC.UBA.AR
Rodrigo Echeveste[4]            RECHEVESTE@SINC.UNL.EDU.AR
Enzo Ferrante[1]                 EFERRANTE@DC.UBA.AR

[1] *Laboratory of Applied Artificial Intelligence, Institute of Computer Sciences, CONICET - Universidad de Buenos Aires, Argentina*

[2] *Weizmann Institute of Science, Rehovot, Israel*

[3] *APOLO Biotech, Buenos Aires, Argentina*

[4] *Research Institute for Signals, Systems and Computational Intelligence, sinc(i), CONICET - Universidad Nacional del Litoral, Argentina*

**Editors:** Accepted for publication at MIDL 2026

## Abstract

In this work, we study uncertainty estimation for anatomical landmark-based segmentation on chest X-rays. Inspired by hybrid neural network architectures that combine standard image convolutional encoders with graph-based generative decoders, and leveraging their variational latent space, we derive two complementary measures: (i) latent uncertainty, captured directly from the learned distribution parameters, and (ii) predictive uncertainty, obtained by generating multiple stochastic output predictions from latent samples. Through controlled corruption experiments we show that both uncertainty measures increase with perturbation severity, reflecting both global and local degradation. We demonstrate that these uncertainty signals can identify unreliable predictions by comparing with manual ground-truth, and support out-of-distribution detection on the CheXmask dataset. More importantly, we release **CheXmask-U** (dataset), a large scale dataset of 657,566 chest X-ray landmark segmentations with per-node uncertainty estimates, enabling researchers to account for spatial variations in segmentation quality when using these anatomical masks. Our findings establish uncertainty estimation as a promising direction to enhance robustness and safe deployment of landmark-based anatomical segmentation methods in chest X-ray. A fully working interactive demo of the method is available at CheXmask-U-demo and the source code at CheXmask-U-code.

**Keywords:** landmark-based anatomical segmentation, uncertainty estimation, graph neural networks, VAE, out-of-distribution detection, chest x-ray

## 1. Introduction

Uncertainty estimation is crucial for the safe deployment of medical segmentation systems. Quantifying model confidence enables clinicians to identify cases where predictions may be unreliable, facilitating appropriate human intervention and improving overall diagnostic reliability (Abdar et al., 2021; Zou et al., 2023).

Traditional pixel-based segmentation approaches rely on convolutional neural networks or Transformer architectures that produce dense masks, typically trained with loss functions

such as cross-entropy or Dice coefficient (Ronneberger et al., 2015) that treat pixels independently, resulting in anatomically implausible predictions that violate structural constraints and exhibit topological inconsistencies, which are critical limitations in clinical applications, where anatomical accuracy is crucial (Bohlender et al., 2023).

Anatomical structures in medical imaging typically present characteristic topologies that remain relatively consistent across individuals, particularly in applications such as chest X-ray analysis where cardiac and pulmonary structures follow predictable spatial relationships. Landmark-based segmentation addresses these topological limitations by representing anatomical structures as graphs of interconnected landmarks, incorporating anatomical constraints by construction and ensuring topological correctness. Following this direction, HybridGNet (Gaggion et al., 2021, 2023a) is a landmark-based segmentation model which combines convolutional neural networks (CNNs) for image feature encoding with graph convolutional networks (GCNNs) for landmark decoding, leveraging variational autoencoders (VAEs) in the graph domain, to enforce anatomical plausibility in the decoded landmarks. Despite the advantages of hybrid architectures for landmark-based segmentation which preserve anatomical topology and incorporate structural constraints, no prior work has addressed uncertainty estimation in such models, representing a significant gap for their practical use.

While large-scale anatomical segmentation datasets like CheXmask (Gaggion et al., 2024) have accelerated research in chest X-ray analysis, they provide only image-level quality assessments, offering limited insight into which anatomical regions are reliably segmented. Fine-grained uncertainty information at the node level would allow users to selectively leverage trustworthy regions, weight contributions from different anatomical structures, and improve robustness in downstream applications. To address this gap, we introduce CheXmask-U, a large-scale dataset of 657,566 chest X-ray landmark segmentations with precomputed per-node uncertainty estimates, enabling spatially informed usage of anatomical masks. The original CheXmask dataset has been used for a variety of applications, like analyzing whether CNNs rely on spurious, non-clinical regions when classifying chest X-rays (Sourget et al., 2025) or exploring counterfactuals in the context of image segmentation (Mehta et al., 2026). We expect that CheXmask-U will further extend these possibilities by enabling localized analyses of uncertainty within the segmentations.

In this work, we study different approaches to produce uncertainty estimates in variational architectures for landmark-based segmentation. Specifically, we focus on lung and heart segmentation in chest X-rays and exploit the VAE latent space to quantify uncertainty in two complementary ways: (i) by analyzing the latent distribution to capture latent-variable–based predictive uncertainty uncertainty, and (ii) through Monte Carlo sampling from the latent posterior, generating multiple stochastic landmark predictions per input. Recent studies have highlighted that uncertainty estimates derived from standard VAEs are not necessarily informative about true data ambiguity. In particular (Catoni et al., 2025) showed that conventional VAEs may produce unreliable uncertainty representations that fail to correlate with perceptual or semantic variability. These findings motivate a closer examination of how latent-space uncertainty behaves in structured medical tasks, such as anatomical landmark localization, where meaningful confidence estimates are critical. To this end, we validate our approach through systematic corruption experiments, including occlusions and Gaussian noise, following the evaluation protocol proposed by (Catoni et al.,

2025), and demonstrate its utility for downstream tasks such as error prediction and out-of-distribution detection.

Our contributions are as follows:

- **Uncertainty Estimation Framework.** We propose a principled approach to quantify uncertainty in landmark-based segmentation using a variational CNN–graph model, capturing both latent-space and predictive uncertainties.

- **Comprehensive Validation.** We validate our uncertainty measures with controlled corruption experiments, showing strong correlation with landmark errors and effectiveness for out-of-distribution detection.

- **CheXmask-U Dataset Release.** We release **CheXmask-U**, a large-scale dataset of 657,566 chest X-ray landmark segmentations with per-node uncertainty estimates, providing a resource to advance uncertainty research in anatomically grounded medical segmentation.

## 2. Related work

### 2.1. Landmark-based segmentation.

A growing body of work has explored landmark-based representations for medical structures, motivated by computational efficiency, reduced annotation requirements, and the ability to capture key anatomical relationships. Earlier shape-based approaches like statistical shape models (Cootes et al., 1995; Heimann and Meinzer, 2009) encoded anatomical priors explicitly through landmark constraints but relied on hand-crafted features and limited flexibility to model large anatomical variability. Recent deep learning methods have made landmark-based segmentation more robust and scalable. Along this direction, HybridGNet (Gaggion et al., 2021, 2023a) demonstrated the effectiveness of combining CNNs and GCNNs for decoding anatomical landmarks, effectively learning structured representations that preserve topology and spatial coherence as opposed to dense pixel-level segmentation models.

Despite these advantages, existing landmark-based frameworks treat landmark predictions deterministically and do not account for uncertainty in landmark positions. This omission limits their reliability in realistic settings, where anatomical structures may be partially occluded, distorted, or poorly visible due to imaging artifacts or pathology. Uncertainty estimation in this context could serve as a mechanism to identify unreliable landmarks, support selective trust in specific regions, and guide downstream decision-making in safety-critical applications.

### 2.2. Uncertainty estimation.

Uncertainty estimation in deep learning encompasses two primary types: *aleatoric uncertainty* (arising from inherent data noise and annotation ambiguity) and *epistemic uncertainty* (stemming from model limitations and insufficient training data (Kendall and Gal, 2017)). In medical applications, both sources contribute to prediction uncertainty and must be carefully accounted for to ensure reliable and trustworthy clinical decision support.

This motivates the need for robust uncertainty quantification (UQ) frameworks that can explicitly assess model confidence and guide interpretation. In this work, we primarily focus on uncertainty derived from latent-variable sampling in a variational model, without marginalizing over model parameters.

A wide range of UQ techniques have been explored for medical image segmentation, predominantly at the pixel level. Bayesian inference provides principled frameworks to quantify model uncertainty but remains from computational prohibitive for large-scale networks. Approximate Bayesian methods such as Monte Carlo dropout (Gal and Ghahramani, 2016) offers an efficient approximation to etimate epistemic uncertainty, while ensembles (Lakshminarayanan et al., 2017) aggregate predictions from multiple independently trained models capturing prediction variability, though at a higher computational cost. Test-time augmentation (TTA) (Ayhan and Berens, 2018) estimates uncertainty instead by evaluating the consistency of predictions under multiple input image perturbations, providing a data-driven view of model reliability. Entropy-based uncertainty estimates have also been explored in pixel-based semantic segmentation, and shown to be highly correlated with erroneous areas (Matzkin et al., 2025; Larrazabal et al., 2021).

Probabilistic segmentation networks have further advanced UQ in medical imaging by explicitly modeling output distributions. Along this line of work, the Probabilistic U-Net (Kohl et al., 2018) combines convolutional decoders with variational inference to represent multiple plausible segmentation hypotheses, while PHiSeg (Baumgartner et al., 2019) introduces hierarchical latent variables to capture uncertainty across spatial scales. These methods enable the generation of diverse yet anatomically consistent segmentation outcomes, effectively separating structured ambiguity from model uncertainty.

However, most existing approaches remain limited to dense, pixel-based formulations. Landmark-based segmentation offers a complementary representation with inherent topological guarantees, yet its uncertainty remains largely unexplored. Extending UQ to the landmark level can improve model transparency by revealing confidence in specific anatomical points and support selective reliance on predictions in downstream clinical workflows.

## 3. Method

### 3.1. HybridGNet Architecture

We employ HybridGNet (Gaggion et al., 2021, 2023a) for landmark-based anatomical segmentation. Each organ ROI is encoded as an anatomical graph, where nodes correspond to landmarks and edges encode anatomical adjacency. Given an image $I$, its segmentation $G = \langle V, A, X \rangle$ is represented by $V$ a fixed set of $M$ nodes representing landmarks, a shared adjacency matrix $A$ encoding anatomical connectivity, and $X \in \mathbb{R}^{M \times 2}$ which contains the 2D spatial coordinates of each landmark that vary across samples.

The network has three main components: (i) *CNN encoder*: $f_e^{\mathcal{I}}(I)$ extracts hierarchical image features and produces a latent representation $z$; (ii) *GCNN decoder*: $f_d^{\mathcal{G}}(z)$ predicts landmark coordinates using graph convolutions and ensuring anatomically plausible predictions via $A$; (iii) *VAE latent space*: $z \sim Q(z|I) = \mathcal{N}(\mu, \sigma^2)$ providing a probabilistic representation of the predicted landmarks (for a detailed architecture description see (Gaggion et al., 2021, 2023a)).

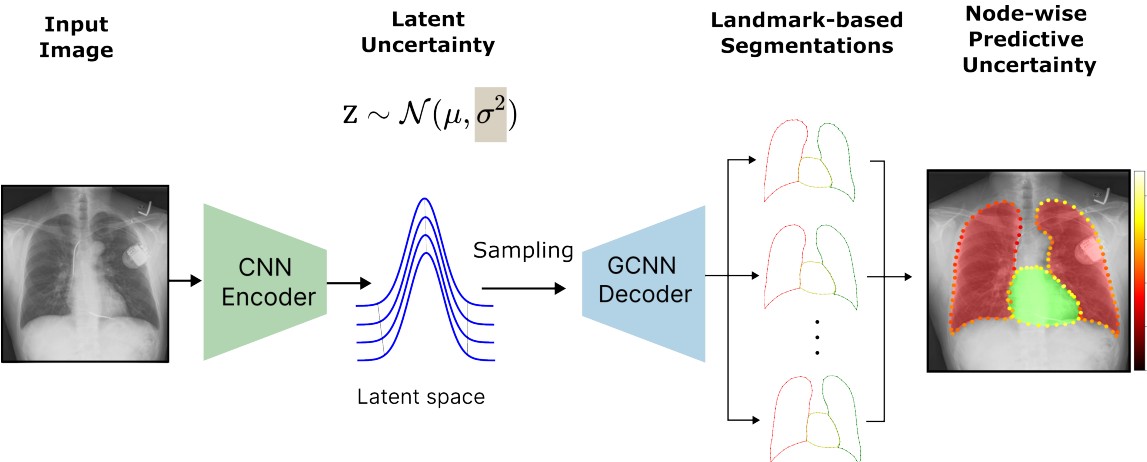

Figure 1: **Proposed uncertainty estimates.** An input image is encoded by the CNN encoder into a probabilistic VAE latent space, producing a *latent uncertainty* $\sigma^2$. Multiple latent samples $\{z^{(i)}\}_{i=1}^{N}$ are drawn from this distribution and decoded through the GCNN, producing multiple output landmark graphs $\{\hat{X}^{(i)}\}_{i=1}^{N}$ from which we can compute the *node-wise predictive uncertainty* as the per-node variance.

Figure 1 shows the overall HybridGNet architecture. We consider two variants of the network. The first uses an image-to-graph skip connections (IGSC) module (Gaggion et al., 2023a), allowing high-resolution image features to flow directly into the GCNN decoder. The second decodes landmarks directly from the CNN latent representation without skip connections. The model is trained to minimize the mean squared error (MSE) between the predicted landmark coordinates $\hat{X}$ and the ground truth $X$, together with a Kullback–Leibler (KL) divergence term that regularizes the VAE latent space.

### 3.2. Uncertainty Estimation

We propose to exploit the VAE latent space of HybridGNet to estimate uncertainty in two complementary ways:

(i) **Latent distribution analysis (latent-variable uncertainty).** Given an encoded latent vector for a single input image, its variance $\sigma^2$ captures model (epistemic) uncertainty regarding landmark placement, providing a global measure of confidence for the predicted anatomical configuration.

(ii) **Node-wise predictive uncertainty via sampling.** To obtain fine-grained uncertainty estimates for each landmark, we perform $N$ stochastic decodings per image. Given an input image $I$, we encode it once to obtain its latent distribution $Q(z|I) = \mathcal{N}(\mu, \sigma^2)$, from which we sample $N$ latent vectors $\{z^{(i)}\}_{i=1}^{N}$ and decode them via the GCNN to generate landmark predictions $\{\hat{X}^{(i)} = f_d^{\mathcal{G}}(z^{(i)})\}_{i=1}^{N}$. The resulting node-wise distributions capture predictive uncertainty, reflecting both model

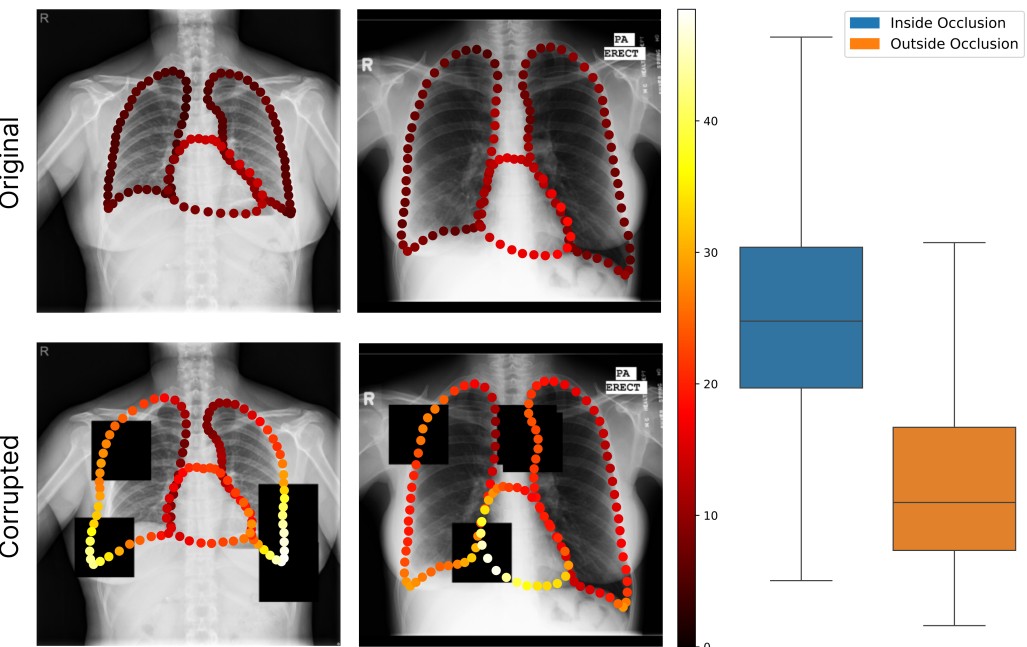

Figure 2: **Node-wise UQ under occlusion.** Two example images are shown alongside their corrupted counterparts with occlusions applied. Node-level uncertainty is visualized with a color gradient, highlighting increased uncertainty in masked regions. The box-plot shows that node-wise uncertainty for nodes inside occlusion blocks is much higher than for nodes outside them.

uncertainty and data ambiguities. This approach is computationally efficient: encoding is performed only once per input image, and the $N$ decodings can be processed in batches, allowing efficient generation of multiple predictions.

## 4. Experiments and Results

### 4.1. Dataset, implementation and training details

We trained a HybridGNet model for lung and heart landmark-based segmentation using the chest X-ray datasets JSRT (Shiraishi et al., 2000), Padchest (Bustos et al., 2020), Montgomery (Candemir et al., 2014) and Shenzhen (Jaeger et al., 2014). Graphs were constructed from the landmark coordinates, with each landmark defining a node and the adjacency matrix encoding anatomical connectivity, shared across all subjects. Following the heterogeneous-label training strategy of (Gaggion et al., 2023b), each batch contains images from a single dataset, and the loss is evaluated only on the nodes corresponding to available annotations. Compared to the original model, we used a higher weighting for the KL divergence term to encourage more structured latent representations: the KL weight was initialized at $1 \times 10^{-5}$ and gradually increased to $1 \times 10^{-2}$ during training. We observed that

very low KL weights lead to near-zero latent posterior variance, resulting in uninformative uncertainty estimates, while excessively large values degrade segmentation accuracy. The adopted KL schedule prevents the decoder from ignoring the latent variables and preserves a structured latent space with stable uncertainty estimates while maintaining predictive performance. On an NVIDIA TITAN Xp GPU, encoding takes 15ms per image for both variants. Decoding is efficient at 7.4-19.7ms (faster without skip connections), as multiple latent samples can be batched together. This makes encoding the main bottleneck, as it is performed only once per image. All experiments are conducted using $N = 50$ stochastic samples, which we found to provide stable uncertainty estimates with a favorable runtime trade-off; sensitivity to $N$ and runtime analysis are reported in Appendix A. We now describe the different experiments that validate our uncertainty quantification approach.

**Validating UQ under Image Occlusions.** We first simulated occlusions under the hypothesis that uncertainty should be higher in occluded areas, applying artificial black-square masks on different image regions. For each image, node-level predictive uncertainty was computed from $N = 50$ stochastic output predictions $\{\hat{X}^{(i)}\}_{i=1}^N$ using the model with skip-connections. As shown in Figure 2, nodes located in the occluded regions exhibit high uncertainty compared to unaffected regions, reflecting the model's ability to localize confidence degradation under partial information loss. Moreover, the boxplot shows uncertainty for nodes falling under occluded and visible areas, confirming significantly larger values for the occluded ones.

**Validating UQ under Noise Corruption.** Following (Catoni et al., 2025), we then validated the UQ under Gaussian noise of increasing intensity, assuming that it should grow as noise increases. Uncertainty was quantified following two approaches: (i) latent-space uncertainty, computed as the average $\sigma$ over the latent distribution, and (ii) node-wise uncertainty from $N = 50$ stochastic output predictions $\{\hat{X}^{(i)}\}_{i=1}^N$, averaged across all nodes. As shown in Figure 3(a), latent-space uncertainty rises with noise levels and then saturates at large corruptions, for both model variants. For the node-wise predictive uncertainty, the no skip-connections model follows this trend. However, the skip-connections variant shows a non-monotonic pattern with a decline at large corruption levels, which may reflect reduced interpretability when skip-connections allow high-resolution features to bypass the variational bottleneck.

**Validating UQ for OOD detection.** We assessed our landmark-based uncertainty measure for out-of-distribution (OOD) chest X-rays detection using the CheXMask dataset (Gaggion et al., 2024), where certain images have been marked as OOD since they correspond to different body parts, views or are very low quality. CheXMask estimated a reverse classification accuracy Dice score (RCA-DSC) per image. Following their analysis, images with RCA-DSC $< 0.7$ were considered as *out-of-distribution* (OOD) and the rest as *in-distribution* (ID). We compute two types of uncertainty-derived scores:

- **Predictive uncertainty score:** we generate 50 stochastic predictions and compute the per-image score as the mean (across nodes) of the node-wise standard deviation, yielding a single predictive-uncertainty scalar for each image.

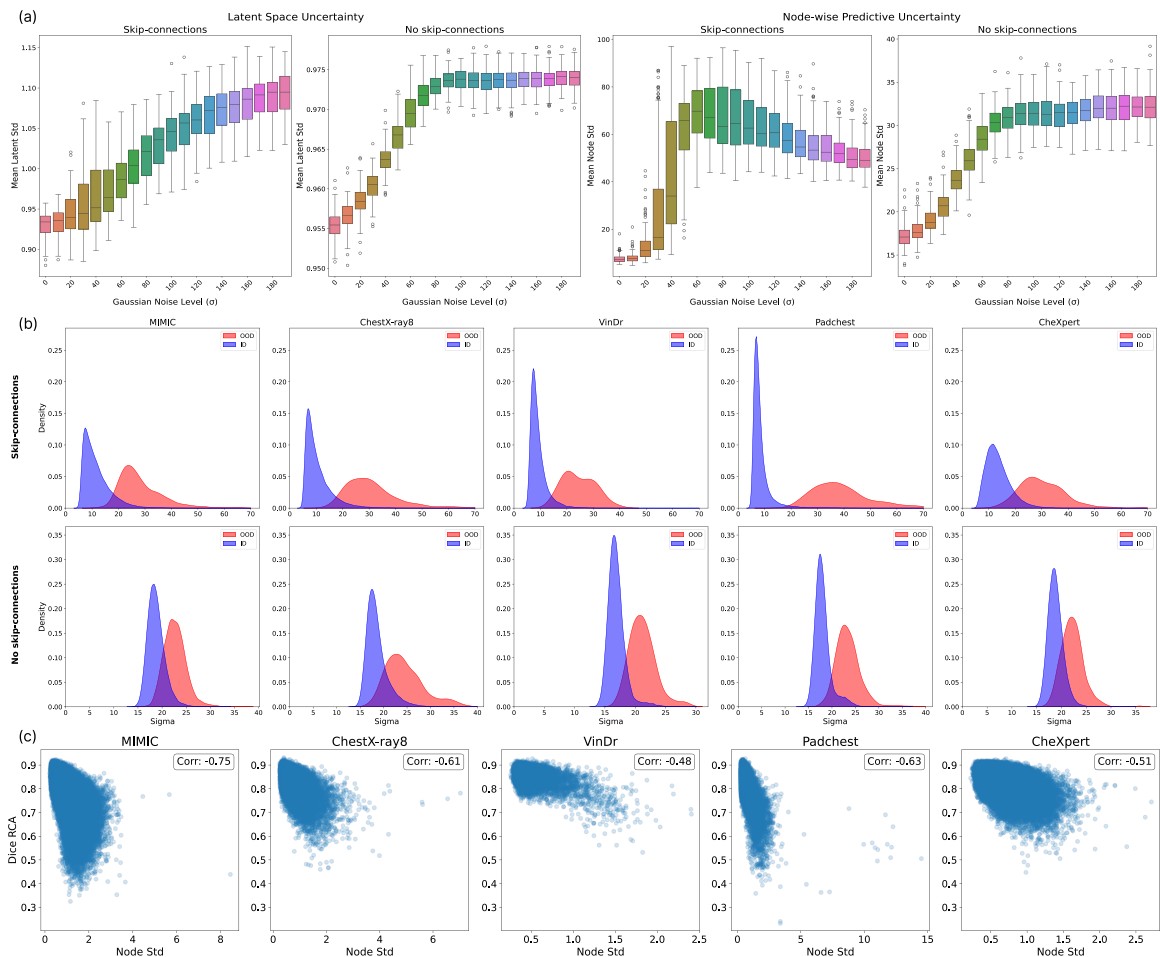

Figure 3: **(a) Uncertainty under Gaussian noise corruption**. Boxplots for models with and without skip-connections show that both latent-space and node-wise predictive uncertainty increase with noise levels, eventually plateauing. **(b) Predictive uncertainty for OOD detection on CheXMask**. KDEs of the per-image uncertainty score show a clear separation between in-distribution and OOD images. **(c) Correlation between uncertainty and segmentation quality.** Across all source datasets from CheXmask-U, higher average uncertainty corresponds to a lower RCA-estimated Dice score, confirming that the measure captures prediction reliability.

- **Latent-space anomaly score:** we use the per-image latent standard deviation $\sigma$ as input to an Isolation Forest (Liu et al., 2008) to compute an anomaly score for each image. We use the `scikit-learn` implementation , performing a train/validation/test split and selecting the best hyperparameter configuration based on validation performance.

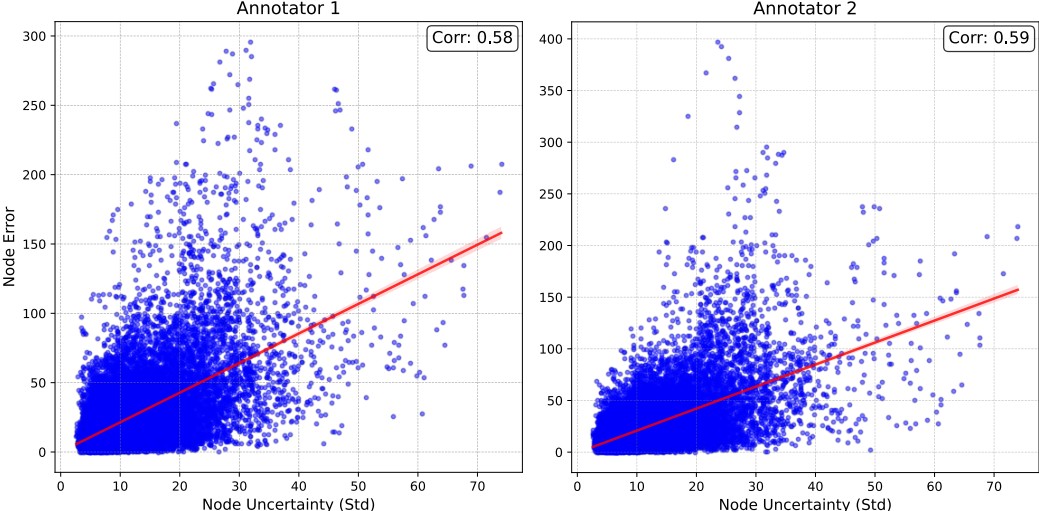

Figure 4: **Correlation between Node Error and Predictive Uncertainty** for annotations generated by 2 experts.

Figure 3(b) presents kernel density estimates (KDE) of the UQ scores for ID vs OOD images, computed separately for each CheXMask source dataset. The top row corresponds to the HybridGNet variant with skip connections, and the bottom row to the plain HybridGNet without skip connections. In all cases the ID and OOD distributions are clearly separable based on the predictive-uncertainty score. Using the predictive-uncertainty score as a scalar OOD detector we obtain an area under the ROC curve (AUC) of **0.98** for the model with skip-connections and **0.93** for the plain model.

For the latent-space approach, Isolation Forest anomaly scores computed on the latent representations yield AUCs of **0.93** for the model with skip-connections and **0.89** for the plain variant. These results indicate that (i) predictive uncertainty derived from sampled landmark outputs is a strong OOD indicator, and (ii) the latent-space features also provide effective OOD signals when used with a standard anomaly detector.

### 4.2. The CheXmask-U Dataset

One of our main contributions is the creation of CheXmask-U [1], a dataset that provides per-node uncertainty estimates for the 657,566 chest X-ray landmark segmentations included in the CheXmask dataset (Gaggion et al., 2024). CheXmask provides anatomical landmark segmentations for frontal chest X-rays across five large-scale datasets: ChestX-ray8 (Wang et al., 2017), CheXpert (Irvin et al., 2019), MIMIC-CXR-JPG (Johnson et al., 2019), Padchest (Bustos et al., 2020), VinDr-CXR (Nguyen et al., 2022). These segmentations are increasingly used by the research community for downstream tasks (Mehta et al., 2026; Sourget et al., 2025). However, CheXmask currently provides only image-level quality assessment via RCA-Dice scores, offering limited insight into which specific anatomical

---

1. https://huggingface.co/datasets/mcosarinsky/CheXmask-U

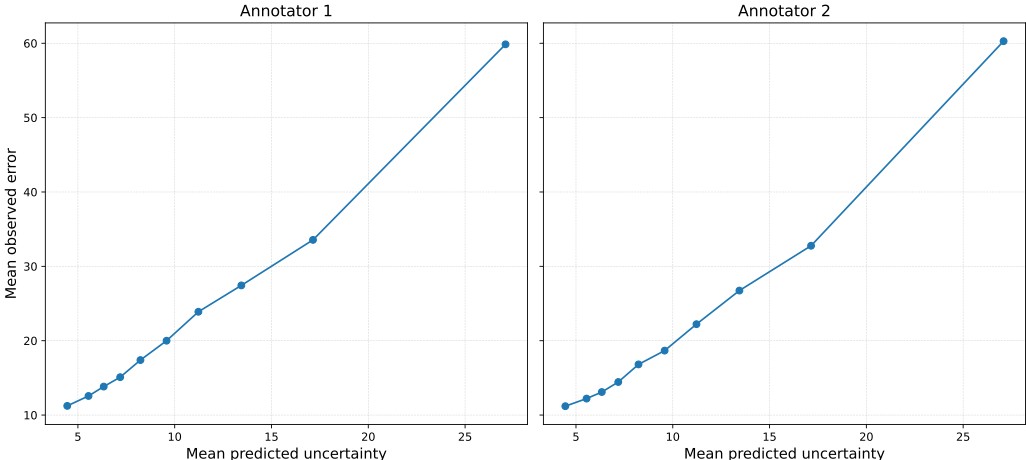

Figure 5: Reliability analysis of node-level uncertainty. Landmark predictions are binned into uncertainty quantiles, and for each bin the mean observed landmark error is plotted against the mean predicted uncertainty with results shown separately for the two annotators.

regions are reliably segmented. CheXmask-U extends this by adding per-node uncertainty estimates, providing fine-grained spatial information about segmentation reliability. This allows users to make informed decisions about which anatomical regions to trust and use in their applications, without requiring access to the segmentation model or computational resources for uncertainty estimation.

**Dataset generation.** The uncertainty estimates were generated using the original HybridGNet model weights (Gaggion et al., 2024), computing $N = 50$ stochastic samples per image for all images included in the original CheXMask. For each node, we report both the mean predicted coordinates and standard deviation, providing fine-grained uncertainty information alongside the anatomical landmark positions.

**Validation.** We validated the per-node predictive uncertainty against actual landmark error, derived from 255 manually annotated images, with two independent annotations. Figure 4 shows a strong positive correlation between predicted uncertainty and actual error, confirming that higher uncertainty reliably indicates less accurate predictions. To further assess the consistency between predicted uncertainty and observed error, we performed a reliability analysis by binning landmark predictions according to their predicted uncertainty. Specifically, nodes were grouped into uncertainty quantiles, and for each bin we computed the mean predicted uncertainty and the corresponding mean landmark error. Figure 5 shows that bins with higher predicted uncertainty exhibit systematically larger observed errors for both annotators.

We further validated CheXmask-U uncertainty estimates by evaluating their relationship with RCA-estimated Dice scores. For each image, the mean node-wise standard deviation was computed as a proxy for overall uncertainty. Figure 3(c) shows a clear anti-

correlation: images with higher predicted uncertainty tend to have lower RCA-DSC. Importantly, CheXmask-U provides node-level granularity that RCA-Dice cannot: an image may have acceptable overall quality but contain specific anatomical regions with high uncertainty—information that is crucial for region-specific downstream applications. By providing both landmark coordinates and per-node uncertainty estimates, CheXmask-U enables researchers to: (i) selectively use landmarks based on confidence thresholds for their specific anatomical regions of interest, (ii) weight contributions from different anatomical regions based on reliability, and (iii) assess segmentation quality at a spatial granularity not available from image-level scores alone. All uncertainty estimates are pre-computed and publicly available, eliminating the need for users to perform uncertainty quantification themselves.

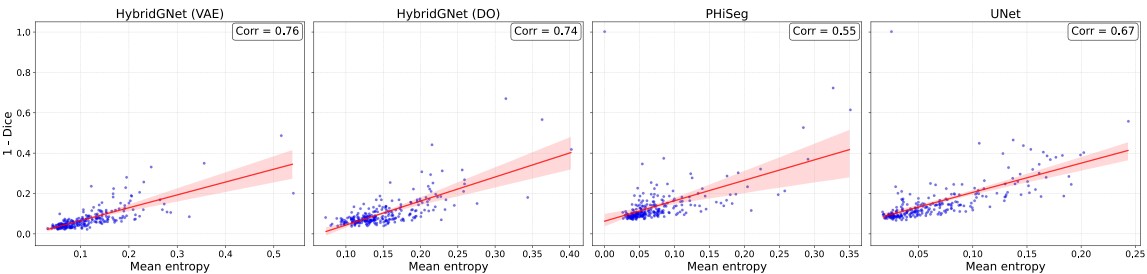

Figure 6: Correlation between predictive uncertainty and segmentation error (1-Dice) for different uncertainty quantification methods at the pixel level. Ground-truth masks are obtained by averaging the annotations from two experts.

### 4.3. Comparison with pixel-level UQ baselines

We compare our landmark-based method against established pixel-level uncertainty quantification methods under a unified evaluation protocol. We consider the following approaches: (i) a pixel-level U-Net segmentation model with Monte Carlo (MC) Dropout, (ii) the probabilistic segmentation model PHiSeg (Baumgartner et al., 2019), (iii) HybridGNet with MC Dropout, and (iv) the proposed variational HybridGNet. For all methods, uncertainty is computed in the same manner: we generate $N = 50$ stochastic pixel-level segmentation masks per image, compute the pixel-wise predictive entropy of the mean segmentation, and average entropy over ground-truth foreground pixels, following (Mehrtash et al., 2020). This yields a single global uncertainty score per image that can be directly compared across methods.

To implement HybridGNet with MC Dropout, we remove the variational latent space and train a deterministic version of HybridGNet with dropout applied to the encoder layers. At inference time, dropout is kept active to produce multiple stochastic predictions. For both HybridGNet variants, landmark-based outputs are deterministically converted into pixel-level segmentation masks (by filling-in the contours) before computing entropy, ensuring a fair comparison with pixel-based methods. Figure 6 reports the correlation between predictive entropy and segmentation error (1 - Dice) across methods. The proposed vari-

ational HybridGNet exhibits a stronger uncertainty–error correlation than the pixel-level baselines, indicating a more reliable global uncertainty signal. This analysis complements the node-level uncertainty evaluation, which provides fine-grained spatial information that cannot be obtained from pixel-based UQ methods alone.

Finally, we note a practical advantage of the proposed approach. Unlike MC Dropout or pixel-level variational models, which require a full forward pass for each stochastic sample, our method encodes the image only once and performs multiple stochastic decodings from the latent space, resulting in significantly lower computational overhead for uncertainty estimation.

## 5. Conclusion

We introduced a framework for uncertainty estimation in landmark-based anatomical segmentation, leveraging the VAE structure of HybridGNet. To our knowledge, this is the first work to provide node-level uncertainty for anatomical landmarks, addressing a gap where prior uncertainty estimation has largely focused on pixel-based segmentation.

Our experiments demonstrate that both latent-space and node-wise predictive uncertainties respond appropriately to image perturbations, including occlusions and Gaussian noise, and serve as effective indicators of out-of-distribution chest X-rays. These validations confirm that landmark-based uncertainty captures meaningful information about prediction reliability. In addition, comparisons with established pixel-level uncertainty methods show that the proposed approach provides complementary, spatially informative signals with lower computational overhead.

We release **CheXmask-U**, a large-scale dataset of 657,566 chest X-ray landmark segmentations with per-node uncertainty, providing a valuable resource for downstream applications that can leverage landmark-level confidence.

Overall, our findings show that uncertainty estimation for landmark-based segmentation is both feasible and valuable, providing a level of interpretability and safety beyond pixel-level approaches. Future work could extend this framework to multi-organ or 3D imaging, further enhancing clinical utility.

## Acknowledgments

EF was supported by the Google Award for Inclusion Research, a Googler Initiated Grant and the Distinguished International Associate award of the Royal Academy of Engineering.

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

## Appendix A. Sensitivity to number of samples

We analyze the sensitivity of the proposed uncertainty estimates to the number of samples used at inference time. We report the correlation between predicted uncertainty and land-mark error, measured against inter-annotator discrepancies from two independent expert annotations, for different values of $N$. Table A shows that uncertainty–error agreement remains stable across a wide range of sample counts, with performance saturating well before $N = 50$. This indicates that the uncertainty estimates are robust to the choice of $N$ and that a moderate number of samples is sufficient in practice.

| Model | $N$ | Annotator 1 | Annotator 2 | Runtime (s) |
|---|---|---|---|---|
| | 10 | 0.52 | 0.52 | 32.2 |
| | 25 | 0.54 | 0.54 | 74.0 |
| HybridGNet (MC Dropout) | 50 | 0.59 | 0.59 | 142.3 |
| | 100 | 0.59 | 0.60 | 281.4 |
| | 200 | 0.59 | 0.59 | 556.8 |
| | 10 | 0.53 | 0.55 | 7.2 |
| | 25 | 0.56 | 0.57 | 8.0 |
| HybridGNet (Variational) | 50 | 0.59 | 0.60 | 9.0 |
| | 100 | 0.60 | 0.60 | 11.5 |
| | 200 | 0.60 | 0.61 | 16.4 |

Table 1: Sensitivity of uncertainty–error correlation to the number of Monte Carlo samples $N$ for variational HybridGNet and HybridGNet with MC Dropout. Correlation is measured between predicted uncertainty and landmark error using inter-annotator discrepancies from two independent expert annotations. Total runtime over all 255 annotations is reported in each case.

