# OpenReview forum: "CheXmask-U: Quantifying uncertainty in landmark-based anatomical segmentation for X-ray images"
_MIDL.io/2026/Conference — MIDL 2026 Poster_

### Official Review · Reviewer_HdQR · 2026-01-07

**Confidence:** 5
**Preliminary Rating:** 2
**Final Rating:** 3

**Summary:**

This paper presents CheXmask-U, an extension of the large-scale CheXmask dataset, providing additional per-landmark uncertainty estimates. A hybrid CNN/GCN variational autoencoder network (HybridGNet) is used to provide both latent-space uncertainty and node-wise uncertainty, obtained from MC sampling the latent distribution. Evaluation is performed by assessing the correlation of the uncertainty estimates under different image corruptions and on OOD detection.

**Strengths:**

* The usage of hybrid architectures with grid-convolutional encoders and graph-convolutional decoders is under-explored and worth investigating.
* A novel way of predicting graph-based anatomical landmark uncertainty is presented.
* The predicted uncertainty is assessed on simulated data corruption and OOD detection and seems to correlate well with landmark error.
* The paper is well-written and easy to follow.

**Weaknesses:**

* While the uncertainty appears empirically useful, the lack of calibration analysis makes it difficult to interpret these values as reliable confidence estimates (see below).
* The authors claim that the proposed HybridGNet approach captures epistemic uncertainty (p. 2, last paragraph). This claim is not supported by the methodology, as VAEs with latent space sampling only capture aleatoric uncertainty (see below).
* No comparison against other UQ methods, such as MC dropout.

**Detailed Comments:**

* The abstract could be made more focused by removing the first two sentences.
* Fig 2 appears to be broken: Only the top-left X-ray is partially shown, the remaining 3 images are blank.
* Typo on page 8, line 4: ~~moremover~~ moreover
* In contrast to the author's claims, the HybridGNet VAE only captures aleatoric uncertainty:
  * Since the proposed uncertainty estimates rely only on latent-variable sampling with point estimates for the model parameters, the approach does not marginalize over model weights and therefore does not capture epistemic uncertainty in the Bayesian sense. The resulting uncertainty primarily reflects aleatoric uncertainty.
  * While this does not invalidate the empirical findings, it creates conceptual confusion for the reader.
* Missing calibration metrics:
  * Usage of commonly accepted calibration metrics would considerably strengthen the paper's claims, such as reliability diagrams (predicted uncertainty vs. observed uncertainty/empirical error), Chi-Squared QQ plots, proper scoring rules (such as NLL), sigma-scaling factor/conformal prediction, etc.
* The release of CheXmask-U dataset is appreciated. However, it is not a strong dataset contribution, as the provided landmark uncertainties are estimates from HybridGNet, containing HybridGNet's architectural priors, are not annotated by humans and are not calibrated.
* Many references only contain "First Author et al.", please fix.

**Justification Of Final Rating:**

The authors have addressed my major concerns, including correcting the incorrect uncertainty claims, adding comparisons to MC dropout baselines, and providing additional reliability diagram analyses. These changes substantially improve the clarity and empirical support of the paper. Thus, I will increase my score.
However, the proposed uncertainty estimates still rely on standard latent-variable sampling, and the released dataset consists of model-derived, uncalibrated uncertainty values. Thus, my final rating is "borderline".

**Justification Of The Preliminary Rating:**

This paper presents a useful approach to uncertainty estimation in landmark-based segmentation and releases an addition to a large-scale dataset of model-derived uncertainty estimates. However, the interpretation of the latent-derived predictive uncertainty as epistemic is not clear. Moreover, the missing calibration analysis and comparison to other baseline UQ methods result in my rating of "weak reject". I'm willing to increase my score if my concerns can be addressed in the rebuttal.

**Questions To Address In The Rebuttal:**

* Please clarify how epistemic uncertainty is captured, or alternatively, correct the wording in the paper.
* How sensitive is the predicted uncertainty to the number of MC samples ($N=50$)?

---

> ### Author Response · Authors · 2026-01-23
>
> We thank the reviewer for highlighting that we introduce “a novel way of predicting graph-based anatomical landmark uncertainty” and that the paper is “well-written and easy to follow”. In what follows, we address the comments raised by the reviewer. Please note that we have updated a new version of the manuscript with additional experiments, where additions are highlighted in blue.
>
> **On the relevance of publicly releasing the CheXmask-U dataset.**
> We thank the reviewer for raising this important point and would like to clarify the intended role of the released uncertainty values. As noted by the reviewer, we agree that the proposed uncertainty estimates are inherently *model-dependent*, as they are produced for segmentations generated by our own model, and are not annotated by humans. Consistent with the original motivation of the CheXMask dataset (Gaggion et al., 2024, Nature Scientific Data), CheXMask-U is released to support *downstream analysis tasks*. Importantly, the original CheXMask annotations themselves are already **model-generated**, as they are derived from HybridGNet predictions, yet they have been widely adopted by the community as a useful silver standard. For example, they have been used to train and evaluate segmentation models (Kalkhof et al., 2025, Medical Image Analysis), as anatomical masks for ROI-based analysis (Sourget et al., 2025, MICCAI Workshop), and in counterfactual image generation frameworks (Mehta et al., 2025, MICCAI).
>
> In this regard, CheXMask-U extends the utility of CheXMask by augmenting these annotations with *localized uncertainty estimates*. These uncertainty scores can be leveraged to identify potentially unreliable regions of the released segmentations, weight landmarks by confidence, or filter samples during training and evaluation, thereby enhancing existing downstream workflows rather than replacing them.
>
> The validation experiments we provide on a subset of manually annotated images are primarily intended to offer empirical evidence that the uncertainty estimates associated with the 657,566 samples are *reasonably trustworthy*. Our goal is not to claim optimal or ground-truth uncertainty estimation, but rather to **quantify the reliability of the model-generated landmark annotations** already present in CheXMask.
>
> Beyond the dataset itself, the paper also introduces a novel approach to estimating uncertainty within a landmark-based segmentation framework and provides empirical validation of this method. However, we do not claim that this uncertainty estimation strategy is universally applicable across all segmentation models.
>
> While model-dependent by construction, the proposed uncertainty estimates remain **practically useful**: they enable downstream users to reason about annotation reliability in the released segmentations, guide confidence-aware learning strategies, and support data curation, even when training or evaluating alternative models.
>
> **On epistemic vs. aleatoric uncertainty.**
> We thank the reviewer for pointing this out and agree with the assessment. Our original wording was imprecise: the proposed HybridGNet VAE captures data-dependent uncertainty induced by latent-variable sampling, but does not marginalize over model parameters and therefore does not capture epistemic uncertainty in the Bayesian sense. In the revised version, we remove references to epistemic uncertainty and consistently refer to the predicted uncertainty as latent-variable–based predictive uncertainty, which avoids conceptual confusion while leaving the empirical findings unchanged.
>
> **On comparison with other UQ methods.**
> We agree that empirical comparison is necessary and have added direct comparisons with MC Dropout baselines, including U-Net with MC Dropout, HybridGNet with MC Dropout, and PHiSeg. Using a unified evaluation based on predictive entropy computed from multiple sampled segmentation masks, we show that our method provides competitive or stronger uncertainty–error correlation. This is now included in *Section 4.3* and *Figure 6* of the revised manuscript.
>
> **On references.**
> We have corrected the references to include full author lists instead of placeholders such as “First Author et al.”
>
> **On the abstract.**
> We thank the reviewer for the suggestion and agree that the abstract can be made more focused. In the revised manuscript, we removed the first two introductory sentences and now begin the abstract directly with the problem setting and contributions.
>
> **On sensitivity to the number of MC samples.**
> We additionally analyze the sensitivity of uncertainty estimates to the number of samples N. We include a table reporting uncertainty–error agreement (measured against inter-annotator landmark discrepancies) for different values of N and observe that results remain stable. This can be found under *Appendix A* of the revised manuscript.

---

> > ### Author Response · Authors · 2026-01-23
> >
> > **On calibration metrics.**
> > We thank the reviewer for this suggestion. In the revised manuscript, we have added a reliability-style analysis to assess the relationship between predicted uncertainty and observed landmark error. Specifically, we include reliability diagrams obtained by binning landmark predictions according to predicted uncertainty and plotting the mean empirical error within each bin. This analysis demonstrates that higher predicted uncertainty consistently corresponds to larger observed errors, providing additional validation of the uncertainty estimates. The new figure is referenced and discussed in *Section 4.2* under *Validation*.

---

### Official Review · Reviewer_xKMj · 2026-01-08

**Confidence:** 4
**Preliminary Rating:** 3
**Final Rating:** 3

**Summary:**

The authors integrate a probabilistic hybrid framework that combines CNNs and a grraph-neural network structure into anatomical landmark segmentation. The used methods is VAE-based and therefore allows for drawing samples in a latent space.
The authors evaluated their method on one publicly available dataset (ChestXMask) and compared segmentation and uncertainty quality.
Overall, I like the idea but there are some shortcomings that I am elaborating further below.

**Strengths:**

First of all, the paper is very nicely structured and easy to follow. The motivation is clear as well as the contribution in the method itself.
Absolutely speaking, the results are also convincing regarding uncertainty quality as the reported correlation between node error and predictive uncertainty.

**Weaknesses:**

I have two main concerns why I am hesitant to accept the paper for the conference. The first one is regarding the provided uncertainty dataset. To my understanding, I don't see how the uncertainty provided in the extended version of the dataset can be useful to anyone using a different model as the uncertainties provided are derived from **your method** and not from the ground truth (such as inter-rater variability). The uncertainty provided is super model-depended, i.e. even if your uncertainty provided has high correlation with the error of your prediction, that doesn't necessary mean that this is automatically a highly ambiguous area absolutely speaking. Other models could theoretically perform well in specific areas that you labeled with high uncertainty.

Moreover, you are not comparing against other baselines even though you list them in your related work. I get that you are using a landmark-based approach and the others operate on a pixel-level. However, it is easy to obtain binary masks from landmarks and therefore a comparison is also not too hard to do and should be done indeed here. There needs to be an empirical justification on why I should your approach instead of other established UQ methods.

**Detailed Comments:**

In Section 1 you only have one subsection (1.1) and nothing else. Either you remove the subsection or you make it more structured

**Justification Of Final Rating:**

I appreciate the authors' efforts and I do believe that some aspects in the manuscript have been improved. However, my main concerns about the utility of the published extended dataset haven't cleared up and I still think that uncertainty estimates that are model-specific are not really useful to the general public. Therefore I will keep my score at Borderline.

**Justification Of The Preliminary Rating:**

My main concern lies in the contribution with the dataset as well as a missing comparison against other methods. I don't see how the dataset can be effectively used in other research questions that involve the use of other UQ models.

**Questions To Address In The Rebuttal:**

Please see the two main points from above.

---

> ### Author Response · Authors · 2026-01-23
>
> We would like to thank the reviewer for their comments and for highlighting that our paper is “ very nicely structured”, “easy to follow”, that it introduces a “clear contribution" and that it presents “convincing results”. In what follows, we address the main concerns raised by the reviewer:
>
> 1. **On the relevance of the provided uncertainty dataset**:
>    We thank the reviewer for raising this important point and would like to clarify the intended role of the released uncertainty values.  As noted by the reviewer, we agree that the proposed uncertainty estimates are inherently model-dependent, as they are produced for segmentations generated by our own model. In this context, we emphasize that our primary contribution is the release of the **CheXMask-U** dataset, which comprises **657,566 landmark-based segmentations with node-wise uncertainty estimates**.
>
>    Consistent with the original motivation of the CheXMask dataset (Gaggion et al., 2024, Nature Scientific Data), CheXMask-U is released to support downstream analysis tasks. Importantly, the original CheXMask annotations themselves are already **model-generated**, as they are derived from HybridGNet predictions, yet they have been widely adopted by the community as a useful silver standard. For example, they have been used to train and evaluate segmentation models (Kalkhof et al., 2025, Medical Image Analysis), as anatomical masks for ROI-based analysis (Sourget et al., 2025, MICCAI Workshop), and in counterfactual image generation frameworks (Mehta et al., 2025, MICCAI).
>
>    In this regard, CheXMask-U extends the utility of CheXMask by augmenting these annotations with localized uncertainty estimates. These uncertainty scores can be leveraged to identify potentially unreliable regions of the released segmentations, weight landmarks by confidence, or filter samples during training and evaluation, thereby enhancing existing downstream workflows rather than replacing them.
>
>    The validation experiments we provide on a subset of manually annotated images are primarily intended to offer empirical evidence that the uncertainty estimates associated with the 657,566 samples are reasonably trustworthy. Our goal is not to claim optimal or ground-truth uncertainty estimation, but rather to **quantify the reliability of the model-generated landmark annotations** already present in CheXMask.
>
>    Beyond the dataset itself, the paper also introduces a novel approach to estimating uncertainty within a landmark-based segmentation framework and provides empirical validation of this method. However, we do not claim that this uncertainty estimation strategy is universally applicable across all segmentation models.
>
>    While model-dependent by construction, the proposed uncertainty estimates remain **practically useful**: they enable downstream users to reason about annotation reliability, guide confidence-aware learning strategies, and support data curation, even when training or evaluating alternative models.
>
> 2. **Comparison to other baselines**:
>    We agree that empirical comparison is necessary and have added direct quantitative comparisons with pixel-level probabilistic baselines (e.g., MC Dropout U-Net and PHiSeg). Using a unified evaluation based on predictive entropy–error correlation computed on sampled segmentation masks, we show that our approach provides a stronger global uncertainty signal while retaining landmark-level interpretability. This analysis is now available in the updated manuscript, *Section 4.3*.
>
> 3. **Single subsection in Section 1**:
>    We have revised the structure of the paper by making *Related Work* a standalone section with two subsections (Landmark-based segmentation and Uncertainty estimation), addressing the issue of having a single subsection under *Section 1*.
>
> **References**:
> (Kalkhof et al., 2025, Medical Image Analysis): https://www.sciencedirect.com/science/article/pii/S1361841525001483
> (Sourget et al., 2025, MICCAI Workshop): https://arxiv.org/pdf/2412.04030
> (Mehta et al, 2025, MICCAI): https://arxiv.org/abs/2506.16213
> (Gaggion et al., 2024, Nature Scientific Data): https://www.nature.com/articles/s41597-024-03358-1

---

> > ### Comment · Reviewer_xKMj · 2026-01-27
> >
> > Dear authors,
> >
> > thank you for your detailed response.
> >
> > I appreciate the additional comparison against pixel-wise probabilistic segmentation networks and the results look good. Here I have a follow-up question about the caption of Figure 6: "Ground-truth masks are obtained by averaging the annotations from two experts". What do you mean here? Taking the mode for each pixel? Or do you do it landmark-based as well?
> >
> > Regarding the dataset: I am unfortunately still not convinced about the utility (please note that I acknowledge your methodological contribution). I am aware of how the "ground truth" is obtained for the original dataset. For your provided uncertainty estimates, it is a bit different though. You need to provide a thorough evaluation of how high uncertainty in the new dataset corresponds to high probability of error and vice versa for low uncertainty, i.e. making it calibrated. If this is not ensured, I don't think that using the uncertainty estimates makes sense for downstream tasks. Please let me know if I am misunderstanding something here.

---

> > > ### Author Response · Authors · 2026-01-27
> > >
> > > Dear reviewer,
> > > Thank you for the follow-up questions and for the careful reading of the revised manuscript. We are happy to clarify both points below.
> > >
> > > **Clarification regarding Figure 6 (“averaging the annotations from two experts”)**
> > > Both expert annotators provide *landmark-based annotations*, not pixel masks. Each annotation consists of a fixed set of corresponding landmarks with 2D coordinates. To obtain a single reference segmentation for the pixel-level comparison in Figure 6, we proceed as follows: for each landmark, we compute the *node-wise average* of the two annotations. The resulting averaged landmark set is then converted into a binary mask by filling the anatomical contours. We will clarify this in the final manuscript.
> > >
> > >
> > > **On the utility and calibration of the uncertainty estimates in CheXmask-U**
> > > We agree that for uncertainty estimates to be useful, they must meaningfully relate to empirical error. This is precisely what motivated the *reliability-style analysis* added in the revised manuscript. As described in *Section 4.2* and now shown in *Figure 5* of the revised manuscript, we bin landmark predictions by predicted uncertainty and compute the mean observed landmark error (against manual annotations) within each bin. The resulting curves show a monotonic increase in empirical error with increasing predicted uncertainty, consistently across both annotators. This provides empirical evidence that higher uncertainty corresponds to a higher probability of error. While we do not claim perfect probabilistic calibration in a strict Bayesian sense, this analysis demonstrates that the uncertainty signal is well ordered and informative for practical use.
> > >
> > > Regarding dataset-wide calibration, it is true that manual ground truth exists only for a limited annotated subset, which necessarily restricts calibration analyses to that subset. The full CheXMask dataset does not include human annotations, which is precisely why releasing uncertainty estimates is valuable: if dense ground truth were available at scale, uncertainty would be far less critical.
> > >
> > > Accordingly, the goal of CheXMask-U is not to provide ground-truth aleatoric uncertainty, but to quantify the *reliability of the model-generated landmark annotations* that are already used by the community, and to enable downstream users to weight, filter, or otherwise reason about annotation quality at *node-level* spatial resolution. The validation we report on the manually annotated subset serves exactly the purpose highlighted by the reviewer: to evaluate whether high uncertainty corresponds to a higher probability of error (and conversely, low uncertainty to a lower probability of error).
> > >
> > > Because manual *ground truth is not available* for the full set of 657,566 segmentations, we perform this evaluation using only the available manual annotations, following a protocol analogous to the one used to validate the estimated Dice values reported in the original CheXMask paper.
> > >
> > > We hope this clarifies the intended scope and utility of the dataset.

---

### Official Review · Reviewer_1BYH · 2026-01-10

**Confidence:** 4
**Preliminary Rating:** 4
**Final Rating:** 4

**Summary:**

The paper proposes a framework for quantifying uncertainty in landmark-based anatomical segmentation of chest X-rays by leveraging the variational latent space of a hybrid CNN-graph neural network. It utilises this structure to derive two complementary measures: latent uncertainty to capture global model confidence, and node-wise predictive uncertainty through stochastic sampling to identify specific unreliable anatomical points. The study's primary contribution is the release of CheXmask-U, a large-scale dataset featuring over 650,000 segmentations with per-node estimates that allow researchers to account for spatial variations in segmentation quality

**Strengths:**

The strengths of the paper lie in its novel approach to uncertainty, rigorous validation and sig ificant contribution to open-source medical data.

1. Landmark-based uncertainty: While uncertainty estimation is common for pixel-level segmentation, this paper addresses a major gap by being the first to provide node-level uncertainty for anatomical landmarks

2. Dual Uncertainty Framework: The authors derive two complementary measures: latent uncertainty (capturing global model confidence) and node-wise predictive uncertainty (localising specific unreliable points). This dual approach allows clinicians to assess both the overall reliability of a scan and the precision of specific anatomical boundaries.

3. Release of the CheXmask-U Dataset: The publication includes the release of a massive dataset containing 657,566 landmark segmentations with pre-computed uncertainty. This is quite valuable to the research community as it provides fine-grained spatial information that allows others to selectively use "trustworthy" landmarks without needing the original model or high-end computational resources.

**Weaknesses:**

Here are some weaknesses

1. Lack of Baselines for Landmark UQ: While the paper mentions pixel-level UQ methods like Monte Carlo dropout and ensembles, it does not provide a direct performance comparison against these established techniques adapted for landmark-based tasks, focusing instead on validating its own variational framework.
2. Hyperparameter Sensitivity: The quality of the latent representation for uncertainty quantification appears highly dependent on training-time regularisation, specifically requiring a carefully tuned and gradually increased weighting of the Kullback–Leibler (KL) divergence term to ensure the latent space remains structured

**Detailed Comments:**

Direct Comparison with Pixel-Level Baselines: While the paper correctly notes its novelty as the first landmark-based uncertainty framework, a minor but impactful addition would be a quantitative comparison against established pixel-level probabilistic models like the Probabilistic U-Net or PHiSeg.

**Justification Of Final Rating:**

The authors responded to my concerns and offered convincing clarifications regarding the issues raised, including the non-monotonic uncertainty signals and comparisons with pixel-based uncertainty. Nevertheless, my score remains unchanged, as the novelty of the proposed uncertainty technique is not sufficiently compelling.

**Justification Of The Preliminary Rating:**

This paper is a weak accept because it addresses a critical gap in medical imaging by being the first to provide node-level uncertainty estimates for landmark-based anatomical segmentation, offering topological guarantees that traditional pixel-based models lack. However, it'll be a good exercise to compare it to pixel-based uncertainty estimates

**Questions To Address In The Rebuttal:**

1. Clarification of the Non-Monotonic Uncertainty Signal: Can you explain the technical reason why the image-to-graph skip connections (IGSC) cause a counterintuitive decline in predictive uncertainty under high Gaussian noise? Does reduced interpretability limit the reliability?
2. Why was there no comparative Analysis with pixel-level baselines? I believe they are quite straightforward to implement.
3. Can you also provide more details on how sensitive the uncertainty estimates are to the KL weighting hyperparameter and how do you ensure the model does not suffer from posterior collapse,  which is a common issue in VAEs

---

> ### Author Response · Authors · 2026-01-23
>
> We thank the reviewer for highlighting that our work introduces a “novel approach to uncertainty, rigorous validation and significant contribution to open-source medical data”. In what follows, we answer to the points raised by the reviewer:
>
> 1.  **Clarification of the Non-Monotonic Uncertainty Signal**:
>     It is indeed true that for very high Gaussian noise, uncertainty diminishes slightly when skip connections are included. In the model without skip connections, all information about the input forcefully needs to pass through the latent variable encoding, but as skip connections are added the model can resort to skip connections to represent local low level features. A higher reliance on skip connections is especially expected for very infrequent datapoints, as a “cheaper” solution in terms of the cost function during training. We note however that uncertainty remains substantially higher than the one reported for in-distribution (low noise) examples (paired statistical test on per-image uncertainty $p<10^{-23}$), so that the model’s capabilities to flag corrupted or out-of-distribution examples remains unaffected. We believe this very minor deviation from optimality will be an acceptable trade-off for many users in the community, looking to avoid blurring effects, which are prevalent in VAEs without skip connections.
>
> 2. **Comparative analysis with baseline methods**:
>    We thank the reviewer for the suggestion. We agree that comparing against pixel-level probabilistic models is valuable, and we have now added this analysis in the revised manuscript.We now compare against a pixel level U-Net segmentation model with MC Dropout, the PHiSeg segmentation model (https://arxiv.org/abs/1906.04045), HybridGNet with MC Dropout, and our variational HybridGNet. For all methods, uncertainty is computed in the same way: we generate multiple stochastic pixel-level segmentation masks, compute the pixel-wise predictive entropy of the mean mask, and average entropy over ground-truth foreground pixels (similar to https://arxiv.org/abs/1911.13273 ). Note that to implement HybridGNet with MC Dropout, we removed the variational backbone and trained a deterministic version of HybridGNet with Dropout in the encoder layers. We then produced multiple masks by outputting several predictions while keeping Dropout on at inference time. Moreover, in order to generate pixel-level segmentation masks for both HybridGNet with MC Dropout and variational HybridGNet, we first output the graph-based segmentations, which are then filled-in to generate the pixel level segmentation.
>
>    Using this unified evaluation, our method shows stronger correlation between predictive entropy and Dice score than pixel-level baselines, indicating a more reliable uncertainty measure. We hypothesize that this is due to the fact that most of the errors in pixel-based segmentation models still occur close to the boundaries, as captured by our model.  Importantly, this global measure complements the node-level uncertainty analysis emphasized in the original paper, which cannot be directly obtained from pixel-based models.
>
>    Finally, we note that our approach also offers a practical advantage: unlike MC dropout or pixel-level VAEs that require a full forward pass for each sample, our model encodes the image only once and performs multiple stochastic decodings, making uncertainty estimation significantly more computationally efficient. These results and discussion are now included in the updated version of the manuscript in *Section 4.3* and *Figure 6*.
>
> 3. **Sensitivity with respect to the KL weight hyperparameter**:
>    We investigated the sensitivity of uncertainty estimates to the KL weighting during development. With very low KL weight, the latent posterior variance is close to zero, resulting in uninformative uncertainty estimates. We therefore select a higher but moderate KL weight that preserves a structured latent space while maintaining predictive performance. This is now discussed in the updated manuscript (*Section 3.1*)

---

> > ### Comment · Reviewer_1BYH · 2026-01-30
> >
> > Hi Authors,
> >
> > Thanks for the detailed comments and response. Though the response to the first point is still not clear. If the model relies heavily on skip connections, does it mean the uncertainty score will not be affected? or its an "acceptable trade off"?
> >
> > "We hypothesize that this is due to the fact that most of the errors in pixel-based segmentation models still occur close to the boundaries, as captured by our model"-- I find this to be true as well.

---

> > > ### Author Response · Authors · 2026-01-30
> > >
> > > Dear Reviewer,
> > > Thank you for the follow-up and for the helpful observation. To clarify, the uncertainty score is still affected when using image-to-graph skip connections, but under very high noise it becomes less strictly monotonic with respect to noise magnitude. We believe this occurs because skip connections allow the model to rely more on local features, partially bypassing the latent bottleneck.
> > > Importantly, however, these skip connections do not provide a deterministic shortcut: even when skip connections are used, the model must first sample the latent space to predict the graph node locations, which in turn determine where image features are extracted from the encoder. As a result, the skip pathways remain stochastic continuing to reflect uncertainty, unlike standard CNN skip connections that are fixed in space.
> > > Consequently, uncertainty remains consistently elevated for heavily corrupted inputs and remains useful for identifying unreliable regions, even if strict monotonic calibration with noise is slightly relaxed at extreme corruption levels. We therefore view this as an acceptable trade-off given the benefits skip connections provide.

---

### Author Rebuttal · Authors · 2026-01-23

**Rebuttal:**

We thank the reviewers for their thoughtful and constructive feedback.

In each comment section, we address the main points raised and summarize the changes made in the revised manuscript, which is attached below as supporting material.

**Supporting Material:**

/attachment/8232c075b28ca50cc0798329e1ca67f663b23b30.pdf

---

### Meta-Review · Area_Chair_vNhz · 2026-02-09

**Recommendation:** Accept (Poster)
**Confidence:** 3

**Metareview:**

The paper proposes a framework for quantifying uncertainty in landmark-based anatomical segmentation of chest X-rays by leveraging the variational latent space of a hybrid CNN-graph neural network. It utilises this structure to derive two complementary measures: latent uncertainty to capture global model confidence, and node-wise predictive uncertainty through stochastic sampling to identify specific unreliable anatomical points. Evaluation is performed by assessing the correlation of the uncertainty estimates under different image corruptions and on OOD detection. Proposed uncertainty estimates rely on standard latent-variable sampling, and the released dataset consists of model-derived, uncalibrated uncertainty values. The paper is well-written. However some concerns in the contribution with the dataset are still lying.

---

### Decision · Program_Chairs · 2026-02-13

Accept (Poster)